# RNA sequence to structure analysis from comprehensive pairwise mutagenesis of multiple self-cleaving ribozymes

Jessica M Roberts[1], James D Beck[2], Tanner B Pollock[3], Devin P Bendixsen[1†], Eric J Hayden[1,2,3]*

[1]Biomolecular Sciences Graduate Programs, Boise State University, Boise, United States; [2]Computing PhD Program, Boise State University, Boise, United States; [3]Department of Biological Science, Boise State University, Boise, United States

*For correspondence: erichayden@boisestate.edu

Present address: †Institute of Genetics and Cancer, University of Edinburgh, Edinburgh, United Kingdom

Competing interest: The authors declare that no competing interests exist.

**Abstract** Self-cleaving ribozymes are RNA molecules that catalyze the cleavage of their own phosphodiester backbones. These ribozymes are found in all domains of life and are also a tool for biotechnical and synthetic biology applications. Self-cleaving ribozymes are also an important model of sequence-to-function relationships for RNA because their small size simplifies synthesis of genetic variants and self-cleaving activity is an accessible readout of the functional consequence of the mutation. Here, we used a high-throughput experimental approach to determine the relative activity for every possible single and double mutant of five self-cleaving ribozymes. From this data, we comprehensively identified non-additive effects between pairs of mutations (epistasis) for all five ribozymes. We analyzed how changes in activity and trends in epistasis map to the ribozyme structures. The variety of structures studied provided opportunities to observe several examples of common structural elements, and the data was collected under identical experimental conditions to enable direct comparison. Heatmap-based visualization of the data revealed patterns indicating structural features of the ribozymes including paired regions, unpaired loops, non-canonical structures, and tertiary structural contacts. The data also revealed signatures of functionally critical nucleotides involved in catalysis. The results demonstrate that the data sets provide structural information similar to chemical or enzymatic probing experiments, but with additional quantitative functional information. The large-scale data sets can be used for models predicting structure and function and for efforts to engineer self-cleaving ribozymes.

## Editor's evaluation

This is a valuable study that provides compelling evidence for important nucleotides in five self-cleaving ribozymes. Epistasis analyses are novel in this field.

## Introduction

Challenges with predicting the functional effects of changing an RNA sequence continues to limit the study and design of RNA molecules. Recently, machine learning approaches have made considerable advancements in predicting an RNA structure from a sequence. However, these approaches rely heavily on crystal structures of RNA molecules and sequence conservation of homologs, both of which are limited for RNA molecules compared to proteins (*Calonaci et al., 2020*; *Townshend et al., 2021*). In addition, describing an RNA molecule as a single structure can be inaccurate, and regulatory elements such as riboswitches demonstrate the importance of an ensemble of structures for an RNA function. It is unclear that predictions based on individual structures alone will be able to predict the

functional effects of mutations with the precision needed for many biotechnical and synthetic biology applications, or to predict disease-associated mutations in RNA molecules (*Halvorsen et al., 2010*). This suggests that new experimental data types might be important for understanding, designing, and manipulating the transcriptome.

Self-cleaving ribozymes provide a useful model to study sequence-structure-function relationships in RNA molecules. Self-cleaving ribozymes are catalytic RNA molecules that cleave their own phosphodiester backbone. They were first discovered in viruses and viroids, but numerous families of self-cleaving ribozymes have since been discovered in all domains of life (*Prody et al., 1986*). The CPEB3 ribozyme, for example, was discovered in the human genome and found to be highly conserved in mammals (*Bendixsen et al., 2021*; *Salehi-Ashtiani et al., 2006*). Other self-cleaving ribozymes, such as the hammerhead and twister ribozymes, are found broadly distributed across eukaryotic and prokaryotic genomes (*Perreault et al., 2011*; *Roth et al., 2014*). The biological roles of ribozymes in different genomes and different genetic contexts remain an active area of investigation (*Jimenez et al., 2015*). In addition to being widespread across the tree of life, self-cleaving ribozymes have also been used for several bioengineering applications (*Liang et al., 2011*; *Peng et al., 2021*; *Wei and Smolke, 2015*; *Zhong et al., 2016*). For example, self-cleaving ribozymes are being combined with aptamers to develop synthetic gene regulatory devices, which have biotechnical and biomedical applications where ligand-dependent control of gene expression is desired (*Kobori et al., 2017*; *Kobori et al., 2015*; *Stifel et al., 2019*; *Townshend et al., 2015*).

The testing of mutational effects in ribozyme sequences has been accelerated by high-throughput experimental approaches. Most self-cleaving ribozymes are fairly small (<200 nt), and genetic variants can be made by chemical synthesis of a single DNA oligonucleotide that is then used as a template for in vitro transcription. The self-cleavage activity of the ribozyme requires a precise three-dimensional structure, and therefore activity can be used as a sensitive indirect readout of native structure. Mutations that disrupt the native structure are detected as reduced activity compared to the unmutated 'wild-type' ribozyme. Several methods have been developed to enable the detection of ribozyme function by high-throughput sequencing of biochemical reactions (*Bendixsen et al., 2019*; *Hayden, 2016*; *Kobori and Yokobayashi, 2016*; *Shen et al., 2021*). For self-cleaving ribozymes, each read from the data reports both the mutations and whether or not that molecule was reacted (cleaved) or unreacted (uncleaved). Therefore, high-throughput sequencing allows numerous genetic variants to be pooled together and still observed hundreds to thousands of times in the data. This provides confidence in the fraction cleaved (FC) for each genetic variant in a given experiment, and genetic variants are compared to determine relative activity (RA). Importantly, the data are internally controlled because both reacted and unreacted molecules are observed, which controls for differences in their abundance due to synthesis steps (chemical DNA synthesis, transcription, reverse-transcription, and PCR).

A common approach to confirm structural interactions in RNA and proteins is through analysis of pairs of mutations (*Dutheil et al., 2010*; *Olson et al., 2014*). In this context, it can be useful to calculate pairwise epistasis, which measures deviations in the mutational effects of double mutants relative to the effects of each individual mutation (assuming an additive model of mutational effects). For example, in the case of a base pair, each single mutation would disrupt the base-pairing interaction, destabilizing the catalytically active RNA structure and reducing activity. However, if two mutants together restore a base pair, the RA of the double mutant would have much higher activity than expected from the additive effects of the individual mutations (positive epistasis). In contrast to paired nucleotides, double mutants at non-paired nucleotides tend to have a more reduced activity than expected from each individual mutation (negative epistasis) (*Bendixsen et al., 2017*; *Li et al., 2016*). In the case of two mutations that create a different base pair (i.e., G-C to A-U), it is known that the stacking with neighboring base pairs is also structurally important, and some base pair substitutions will not be equivalent in a given structural context. This creates a range of possible epistatic effects even for two mutations at paired nucleotide positions. In addition, some non-canonical base interactions within tertiary contacts may also show epistasis even when they do not involve Watson-Crick or GU wobble base-pairing interactions. Nevertheless, the propensity for positive epistasis between physically interacting nucleotides suggests that a comprehensive evaluation of pairwise mutational effects should contain considerable structural information.

Here, we report comprehensive analysis of mutational effects for all single and double mutants for five different self-cleaving ribozymes. RA effects of all single and double mutations were determined by high-throughput sequencing of co-transcriptional self-cleavage reactions, and this data was used to calculate epistasis between pairs of mutations. The ribozymes studied include a mammalian CPEB3 ribozyme, a hepatitis delta virus (HDV) ribozyme, a twister ribozyme from *Oryza sativa*, a hairpin ribozyme derived from the satellite RNA from tobacco ringspot virus, and a hammerhead ribozyme (*Bendixsen et al., 2021*; *Burke and Greathouse, 2005*; *Chadalavada et al., 2007*; *Liu et al., 2014*; *Müller et al., 2012*). For each reference ribozyme, a single DNA oligo template library was synthesized with 97% wild-type nucleotides at each position, and 1% of each of the three other nucleotides. This mutagenesis strategy was expected to produce all possible single and double mutants, as well as a random sampling of combinations of three or more mutations. The mutagenized templates were transcribed in vitro, all under identical conditions, where active ribozymes had the opportunity to self-cleave co-transcriptionally. All ribozyme constructs studied cleave near the 5′-end of the RNA, and a template switching reverse transcription protocol was used to append a common primer binding site to both cleaved and uncleaved molecules. Subsequently, low-cycle PCR was used to add indexed Illumina adapters for high-throughput sequencing. Each mutagenized ribozyme template was transcribed separately and in triplicate, and amplified with unique indexes so that all replicates could be pooled and sequenced together on an Illumina sequencer. The sequencing data was then used to count the number of times each unique sequence was observed as cleaved or uncleaved, and this data was used to calculate the FC. The FC of single and double mutants was normalized to the unmutated reference sequence to determine RA. The RA values of the single and double mutants were used to calculate all possible pairwise epistatic interactions in all five ribozymes. We mapped epistasis values to each ribozyme structure to evaluate correlations between structural elements and patterns of pairwise epistasis values. The results indicated that structural features of the ribozymes are revealed in the data, suggesting that these data sets will be useful for developing models for predicting sequence-structure-function relationships in RNA molecules.

## Results and discussion

### Epistatic effects in paired nucleotide positions show stability-dependent signatures

To evaluate how the effects of mutations mapped to the ribozyme structures, we plotted the RA values as heatmaps, similar to previous publications by others (*Andreasson et al., 2020*; *Kobori and Yokobayashi, 2016*; *Figure 1*, *Figure 2*, *Figure 3*, *Figure 4*, *Figure 5*, panel A, large plot). We then used this data to calculate epistasis between pairs of mutations. We first inspected nucleotide positions known to be involved in base-paired regions of the secondary structure of each ribozyme. In this heatmap layout, many paired regions showed an anti-diagonal line of high-activity double mutant variants with strong positive epistasis (*Figures 1–5*, insets, red to blue plots). In addition, pairs of mutations off the anti-diagonal tended to show negative or non-positive epistasis. Pseudo-knot elements that involve Watson-Crick base pairs also showed this pattern, including the single base pair T1 element in CPEB3 (*Figure 1*) and the two base pair T1 element in HDV (*Figure 2*). The layout of mutations in the heatmap places paired nucleotide positions along the anti-diagonal and compensatory double mutants that change one Watson-Crick base pair to another are found on this anti-diagonal. Individual mutations that break a base pair will often reduce ribozyme activity, but the activity can be restored by a second compensatory mutation resulting in positive epistasis. In contrast, double mutants off-diagonal usually disrupt two base pairs (unless they result in a GU wobble base pair). It is expected that breaking two base pairs in the same paired region would be more deleterious to ribozyme activity than breaking one base pair. The epistasis data indicates that two non-compensatory mutations in the same paired region are more deleterious than expected from an additive assumption, and frequently create negative epistasis off-diagonal within paired regions.

To further evaluate epistasis within base-paired regions, we separated epistasis data into three categories based on the number of base pairs that the mutations disrupt. For each ribozyme, we plotted the distribution of epistasis values as violin plots (*Figures 1–5*, panel C). For all ribozymes, the analysis revealed a clear trend. On average, disrupting two base pairs resulted in negative epistasis (mean of distribution), disrupting one base pair shifted the distribution toward more positive epistasis

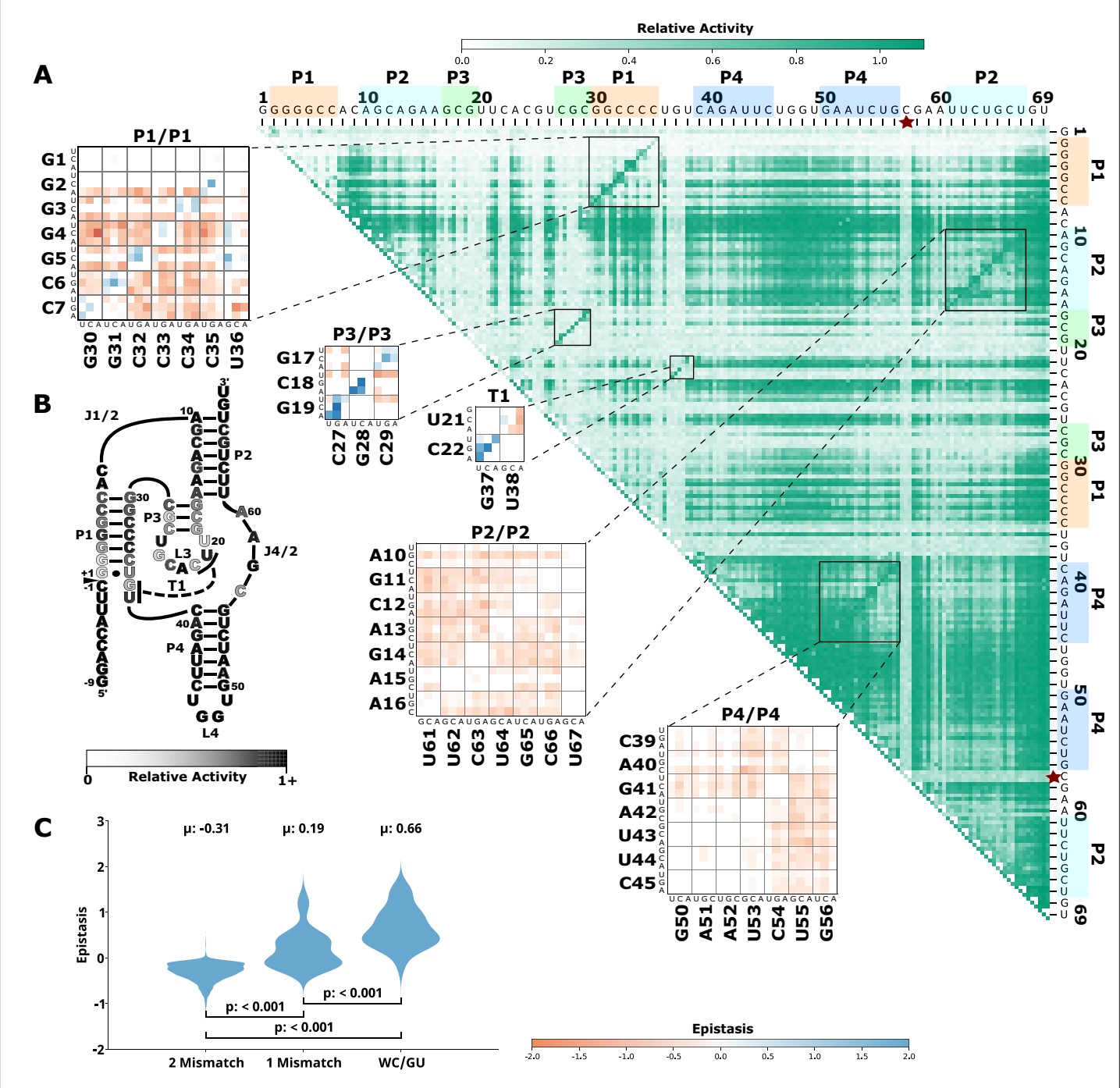

**Figure 1.** Effects of mutations and pairwise epistasis in a CPEB3 ribozyme. (**A**) Relative activity heatmap depicting all possible pairwise effects of mutations on the cleavage activity of a mammalian CPEB3 ribozyme. Base-paired regions P1, P2, P3, P4, and T1 are highlighted and color coordinated along the axes, and surrounded by black squares within the heatmap. Pairwise epistasis interactions observed for each paired regions are each shown as expanded insets for easy identification of the specific epistatic effects measured for each pair of mutations. Instances of positive epistasis are shaded blue, and negative epistasis is shaded red, with higher color intensity indicating a greater magnitude of epistasis. Catalytic residues are indicated by stars along the axes (**A** is reproduced from Figure 1B from *Beck et al., 2022*). (**B**) Secondary structure of the CPEB3 ribozyme used in this study. Each nucleotide is shaded to indicate the average relative cleavage activity of all single mutations at that position. (**C**) Distributions of epistasis values in the paired regions of the CPEB3 ribozyme. Data were categorized as double mutations that result in two mismatches (2 Mismatch), a single mismatch (1 Mismatch), or no mismatches because of a new Watson-Crick base pair or GU wobble results (WC/GU).

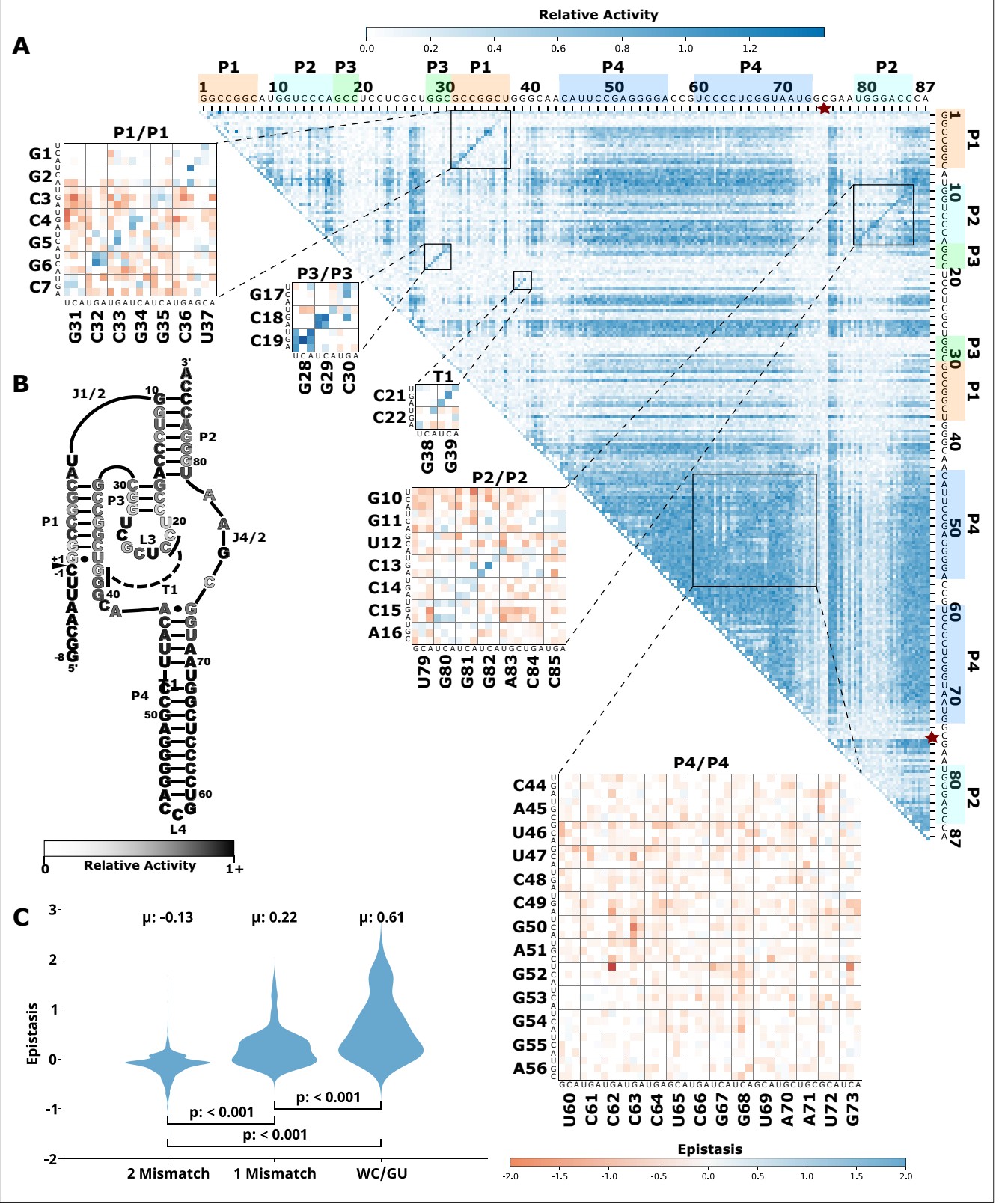

**Figure 2.** Effects of mutations and pairwise epistasis in a HDV self-cleaving ribozyme. (**A**) Relative activity heatmap depicting all possible pairwise effects of mutations on the cleavage activity of an HDV ribozyme. Base-paired regions P1, P2, P3, P4, and T1 are highlighted and color coordinated along the axes, and surrounded by black squares within the heatmap. Pairwise epistasis interactions observed for each paired regions are each shown as expanded insets for easy identification of the specific epistatic effects measured for each pair of mutations. Instances of positive epistasis are shaded

*Figure 2 continued*

blue, and negative epistasis is shaded red, with higher color intensity indicating a greater magnitude of epistasis. Catalytic residues are indicated by stars along the axes. (**B**) Secondary structure of the HDV ribozyme used in this study. Each nucleotide is shaded to indicate the average relative cleavage activity of all single mutations at that position. (**C**) Distributions of epistasis values in the paired regions of the HDV ribozyme. Data were categorized as double mutations that result in two mismatches (2 Mismatch), a single mismatch (1 Mismatch), or no mismatches because of a new Watson-Crick base pair or GU wobble results (WC/GU). HDV, hepatitis delta virus.

values, and the highest epistasis values (mean and max) were found for double mutants that result in zero disrupted base pairs because the two mutations together create a new Watson-Crick or GU Wobble pair. This trend was observed for paired regions in every ribozyme, and in all cases the distributions were significantly different (p<0.05–0.001, Mann-Whitney U test). This pattern of epistasis in paired regions demonstrates the potential for identifying base-paired regions in RNA structures using comprehensive double-mutant activity data.

To further evaluate the potential of epistasis data to identify base-paired regions, we analyzed the epistasis values for each paired region individually. For this analysis, we separated the epistasis values calculated for double mutants that result in a Watson-Crick base pair ('on-diagonal' in heatmaps) from all other double mutants ('off-diagonal' in heatmaps) in each paired region (*Figure 6*). Short-paired regions showed the largest differences in the distributions of epistatic effects for on-diagonal and off-diagonal double mutants, while longer-paired regions showed small differences in these distributions. For example, short-paired regions P3 in CPEB3 and HDV (3 bp), and T1 in the twister (4 bp) showed very large differences in the mean of the distributions. These small regions were highly sensitive to individual mutations, and most pairs of mutations within this region resulted in almost no detectable activity except when they created a different Watson-Crick base pair, leading to the large positive epistasis values (*Figures 1–5*). In addition, in these short-paired regions, we do not see strong negative epistasis. It appears that the strong deleterious effect of a single mutation in these short regions makes a second mutation no more disruptive to activity, resulting in a mean of the distribution near zero for double mutants off-diagonal. In contrast, the largest paired region (HDV P4, 14 bp) showed a very small difference between the distribution of epistasis values found on-diagonal and off-diagonal. This can be rationalized because losing one base pair was not deleterious to the HDV ribozyme activity under our experimental conditions (*Figure 2*), and this does not allow for positive epistasis upon a second mutation. Even the loss of two base pairs in P4 was somewhat tolerated, leading to very little negative epistasis for two mutations at unpaired positions. Taken together, the results are consistent with other observations in both RNA and proteins, where it has been observed that the effects of mutations, and their additivity, have been shown to be dependent on the local thermodynamics of the structured region (*Kraut et al., 2003*; *Moody and Bevilacqua, 2003*).

To explicitly investigate the influence of thermodynamic stability on mutational effects in the data, we calculated the minimum free energy for each paired region and compared mutational effects. We split each paired region into two separate RNA sequences that contained only the base-paired nucleotides, eliminating loop nucleotides, and used nearest neighbor rules to calculate the minimum free energy of their interaction (NUPACK). This approach neglects thermodynamic contributions from terminal loops, but allowed for a consistent approach to compare internal and terminal paired regions. We found a significant negative correlation between the median deleterious effects of single mutations and the minimum free energy of the paired regions (*Figure 6—figure supplement 1*). Clearly, though, thermodynamic stability alone does not explain every mutational effect. For example, CPEB3 P1 is more sensitive to mutations than CPEB3 P2 or P4 even though the latter are less stable. This is likely because P1 is immediately adjacent to the site of self-cleavage, while P2 and P4 are not. Overall, this analysis of thermodynamic stability indicates that for RNA's with unknown structures, more stable structural elements may be harder to identify from epistatic effects alone when there is not a strong deleterious effect of individual mutations. However, it is also possible that more stable elements would show stronger epistasis under different experimental conditions, such as different temperatures or magnesium concentrations (*Peri et al., 2022*).

## Catalytic residues do not have any high-activity mutants

Self-cleaving ribozymes often utilize a concerted acid-base catalysis mechanism where specific nucleobases act as proton donors (acid) or acceptors (base) (*Jimenez et al., 2015*), and mutations at these

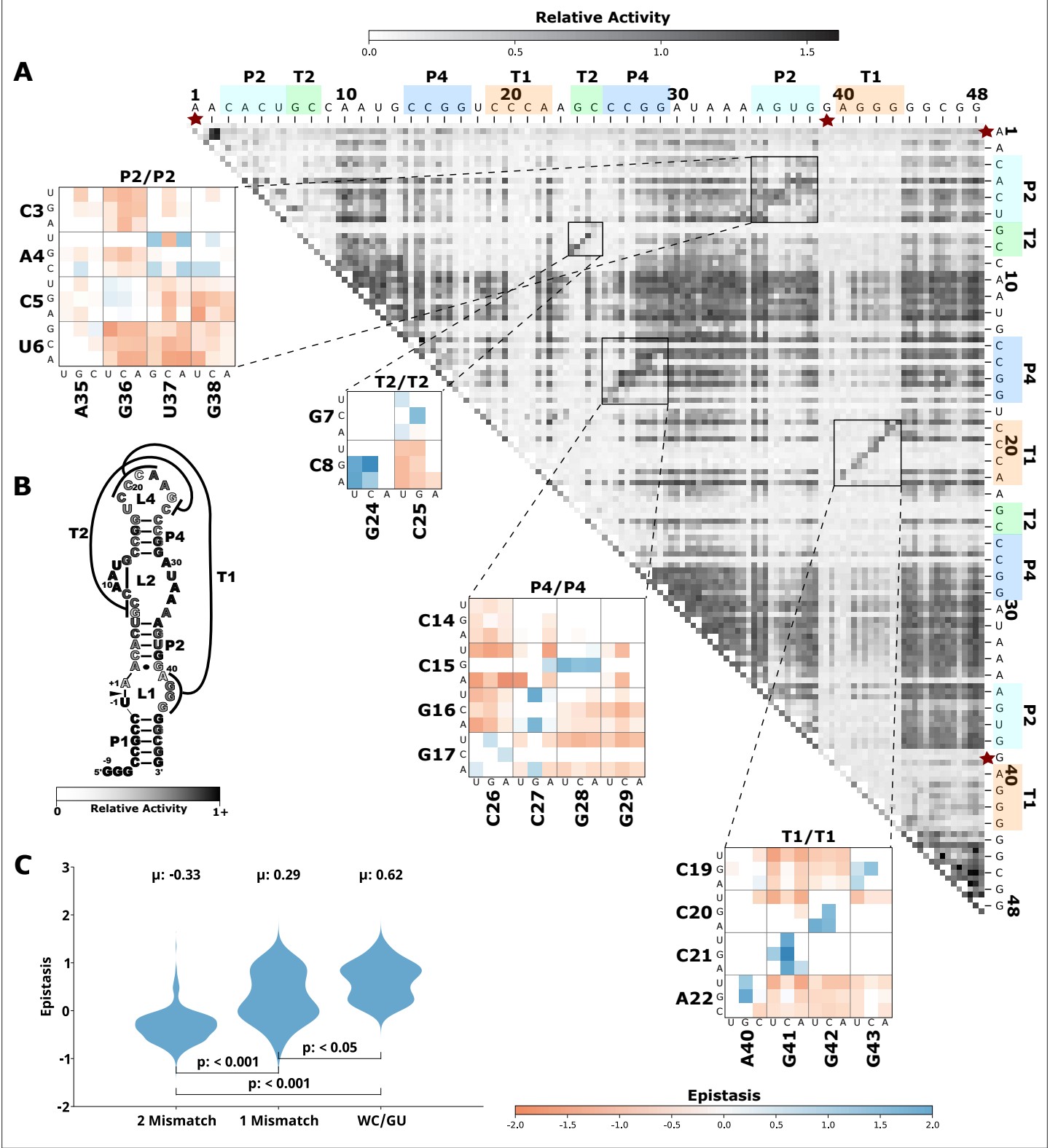

**Figure 3.** Effects of mutations and pairwise epistasis in a twister self-cleaving ribozyme. (**A**) Relative activity heatmap depicting all possible pairwise effects of mutations on the cleavage activity of a twister ribozyme. Base-paired regions P2, P4, T1, and T2 are highlighted and color coordinated along the axes, and surrounded by black squares within the heatmap. Pairwise epistasis interactions observed for each paired region are each shown as expanded insets for easy identification of the specific epistatic effects measured for each pair of mutations. Instances of positive epistasis are shaded blue, and negative epistasis is shaded red, with higher color intensity indicating a greater magnitude of epistasis. Catalytic residues are indicated

eLife Research article    Biochemistry and Chemical Biology | Structural Biology and Molecular Biophysics

*Figure 3 continued*

by stars along the axes. (**B**) Secondary structure of the twister ribozyme used in this study. Each nucleotide is shaded to indicate the average relative cleavage activity of all single mutations at that position. (**C**) Distributions of epistasis values in the paired regions of the twister ribozyme. Data were categorized as double mutations that result in two mismatches (2 Mismatch), a single mismatch (1 Mismatch), or no mismatches because of a new Watson-Crick base pair or GU wobble results (WC/GU).

positions abolish activity. Analyzing the effects of individual mutations will not distinguish catalytic nucleotides from structurally important nucleotides. Comprehensive pairwise mutations, on the other hand, can potentially distinguish between catalytic residues that cannot be rescued by a second mutation, and structurally important nucleotides that can be rescued (positive epistasis). The catalytic cytosines of the CPEB3 (C57) and HDV (C75) act as proton donors due to perturbed pKa values (*Nakano et al., 2000*; *Skilandat et al., 2016*). For the twister ribozyme (*Figure 3*), the guanosine at position G39 acts as a general base, and the adenosine at position A1 acts as a general acid (*Wilson et al., 2016*). The catalytic nucleotides for the Hammerhead ribozyme (*Figure 5*) are the Guanosines located at positions G25 and G39 (*Scott et al., 2013*). The hairpin ribozyme (*Figure 4*) contains catalytic nucleotides at positions G29 and A59 (*Wilson et al., 2006*). In the RA heatmaps, the columns and rows associated with these nucleotides result in low activity values (*Figures 1–5*, *Figure 6—figure supplement 2*). It is important to note that because there is complete coverage of all double mutants in this data set, we can be certain that there are no possible compensatory mutations. These results show how catalytic residues can be identified in the comprehensive pairwise mutagenesis data.

## Unpaired nucleotides show mutational effects that depend on tertiary structure

Ribozymes with mutations to nucleotides found in terminal loops that are not involved in tertiary structure elements showed high RA for most single and double mutants, and essentially no epistasis. This is not surprising if these loops reside on the periphery of the ribozyme and are not involved in structural contacts with other nucleotides. This is the case for L4 of CPEB3 (*Figure 1*), L4 of HDV (*Figure 2*), and L1 and L3 of the hairpin ribozyme (*Figure 4*). Two mutations within these loops do not reduce activity, and mutations in these loops do not rescue other deleterious mutations such as those that break a base pair.

The internal loops LA and LB of the hairpin ribozyme are structurally important (*Figure 4*). Interactions between nucleotides within LB include six non-Watson-Crick base-pairing interactions that are important for the formation of an active ribozyme structure (*Figure 4—figure supplement 1*). Several non-canonical base-base and sugar-base hydrogen bonds between nucleotides within LA are also important for the formation of the active site (*Fedor, 2000*; *Wilson et al., 2006*). Docking between LA and LB is necessary for the formation of a catalytically active ribozyme and is facilitated by a Watson-Crick base pair between nucleotides numbered G1 and C46 in the version of the ribozyme used here (*Rupert and Ferré-D'Amaré, 2001*). In contrast to terminal loop regions, most single mutations within LA and LB resulted in low self-cleavage activity in our data (*Figure 4*). In addition, the double mutants within and between loop A and loop B show several instances of strong positive epistasis (*Figure 4—figure supplement 1C*), and the distributions of epistasis within and between these loops are significantly different than the terminal loops that are not structurally important (*Figure 4D*). This positive epistasis indicates that many of the important structural contacts can be achieved by other specific pairs of nucleotides. For example, the double mutant G1C and C46G shows strong epistasis suggesting that swapping a C-G base pair for the G-C base pair can restore activity by facilitating docking between the two loops. Several double mutants at positions that form non-canonical interactions in LB show positive epistasis. For example, mutation A41G shows positive epistasis when the interacting nucleotide C65 is mutated to a G or U. The non-canonical A45:A59 interaction shows positive epistasis for several pairs of mutations (A45U A59C, A45C A59C, and A45G A59U). Finally, the non-canonical base pair A47:G57 in LB, shows positive epistasis for the A47U:G57A double mutant. The difference between terminal loops and loops with structural importance highlights how activity-based data can help identify non-canonical structures that are challenging to predict computationally, and that might be difficult to identify by other common approaches, such as chemical probing experiments (*Walter et al., 2000*).

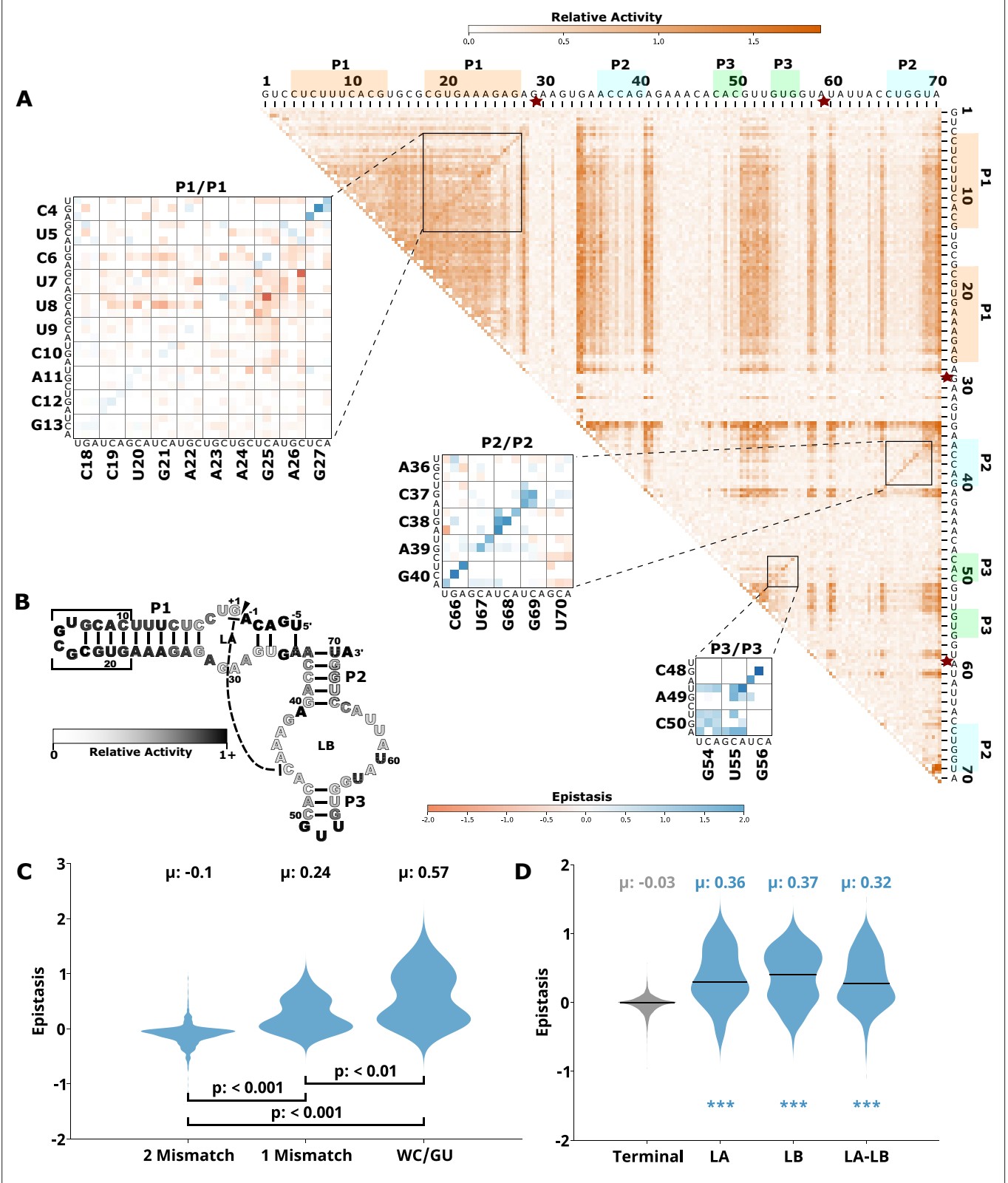

**Figure 4.** Effects of mutations and pairwise epistasis in a hairpin self-cleaving ribozyme. (**A**) Relative activity heatmap depicting all possible pairwise effects of mutations on the cleavage activity of a hairpin ribozyme. Base-paired regions P1, P2, and P3 are highlighted and color coordinated along the axes, and surrounded by black squares within the heatmap. Pairwise epistasis interactions observed for each paired region are each shown as expanded insets for easy identification of the specific epistatic effects measured for each pair of mutations. Instances of positive epistasis are shaded blue, and

*Figure 4 continued on next page*

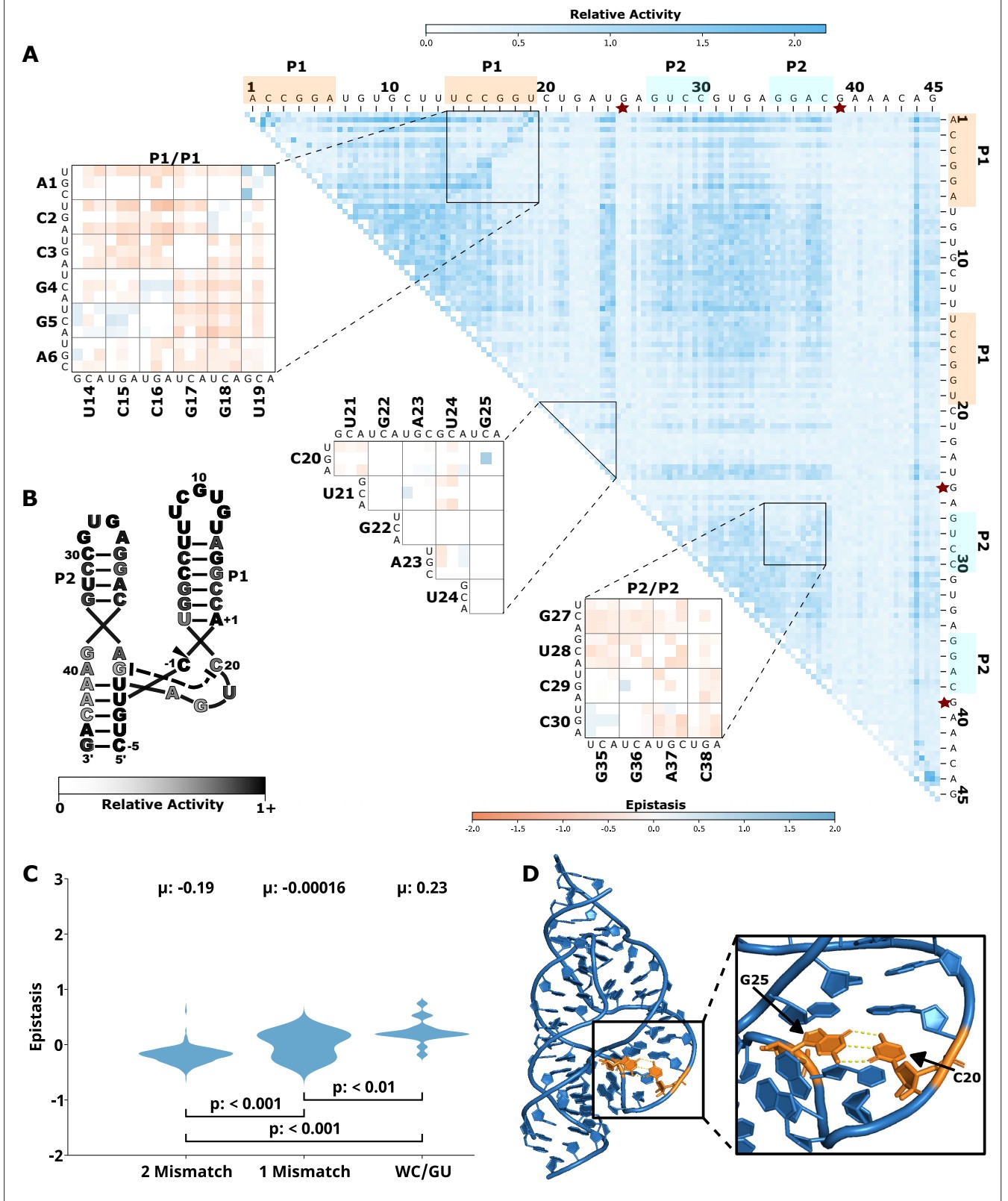

**Figure 5.** Effects of mutations and pairwise epistasis in a hammerhead self-cleaving ribozyme. (**A**) Relative activity heatmap depicting all possible pairwise effects of mutations on the cleavage activity of a hammerhead ribozyme. Base-paired regions, P1 and P2, are highlighted and color coordinated along the axes, and surrounded by black squares within the heatmap. Pairwise epistasis interactions observed for each paired region are each shown as expanded insets for easy identification of the specific epistatic effects measured for each pair of mutations. Instances of positive epistasis

*Figure 5 continued on next page*

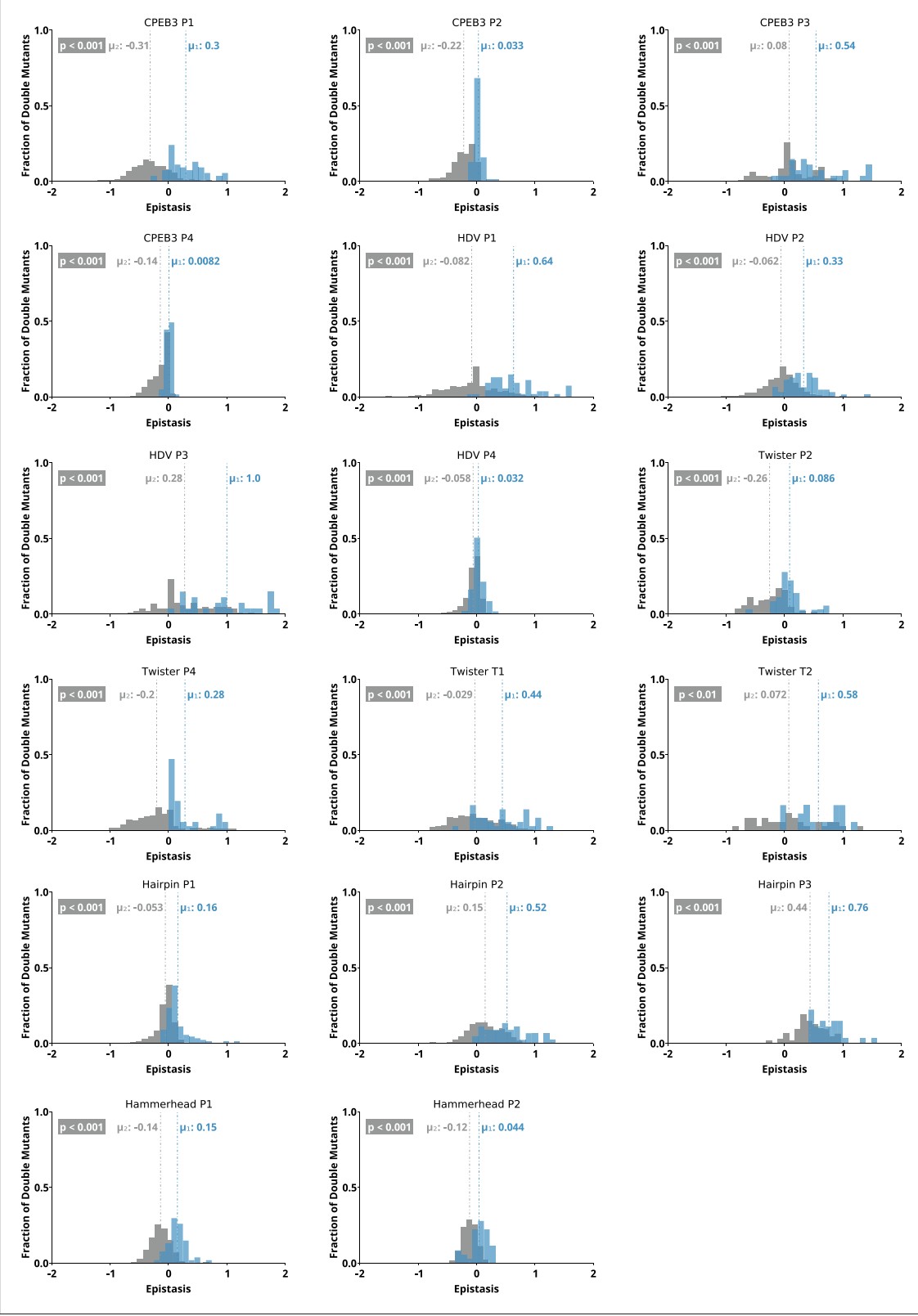

**Figure 6.** Distributions of epistasis values calculated for individual paired regions in all five ribozymes. For each region, epistasis values were separated into double mutants that restore a Watson-Crick base pair ('on-diagonal', blue) and all other double mutants ('off-diagonal', gray). The mean of each distribution (μ) is reported and indicated by the dashed line. The p value is the probability that values were drawn from the same distribution by chance (Mann-Whitney U test).

*Figure 6 continued on next page*

*Figure 4 continued*

negative epistasis is shaded red, with higher color intensity indicating a greater magnitude of epistasis. Catalytic residues are indicated by stars along the axes. (**B**) Secondary structure of the hairpin ribozyme used in this study. Each nucleotide is shaded to indicate the average relative cleavage activity of all single mutations at that position. (**C**) Distributions of epistasis values in the paired regions of the hairpin ribozyme. Data were categorized as double mutations that result in two mismatches (2 Mismatch), a single mismatch (1 Mismatch), or no mismatches because of a new Watson-Crick base pair or GU wobble results (WC/GU). (**D**) The distributions of epistasis values in all terminal stem loops across all five ribozymes, and epistasis observed within loop A, loop B, and between loop A and loop B in the hairpin ribozyme.

The online version of this article includes the following figure supplement(s) for figure 4:

**Figure supplement 1.** Epistasis in the internal loops of a hairpin ribozyme.

Another example of structurally important unpaired regions can be found in the CUGA uridine turn (U-turn) motif in the hammerhead ribozyme (*Figure 5*). This CUGA turn forms the catalytic pocket and positions a catalytic cytosine (-1C) at the cleavage site (*Doudna, 1995*). Crystal structures of the sTRSV ribozyme show a base pair between the nucleotides corresponding to the nucleotides numbered here as C20 and G25 in the ribozyme construct used for our experiments (*Chi et al., 2008*; *Martick and Scott, 2006*). These two nucleotides showed strong positive epistasis for the mutations C20G and G25C, which substitutes a G:C base pair for the original C:G base pair. All other single and double mutants in this region showed low activity, and no instances of strong positive epistasis within or between this motif (*Figure 5*). The low activity resulting from mutations in this region confirms the functional importance of this motif, and indicates that this motif cannot be easily formed or rescued by sequences with up to two mutational differences, except for the G:C base pair swap.

Tertiary interactions between loops in the hammerhead ribozyme provide another example of structurally important loop regions. Type III hammerhead ribozymes, like the one used in this study, contain tertiary interactions between nucleotides in the loops of P1 and P2 that are implicated in structural organization of the catalytic core. A crystal structure of this loop-loop interaction showed a network of interhelical non-canonical base pairs and stacks, with several nucleobases in stem-loop I interacting with more than one nucleobase in stem-loop II (*Chi et al., 2008*; *Martick and Scott, 2006*). However, there are numerous different loop sequences in naturally occurring hammerhead ribozymes indicating that this loop-loop interaction can be formed by a variety of different sequences (*Burke and Greathouse, 2005*; *Perreault et al., 2011*). We therefore anticipated that we would observe a significant level of positive epistasis between these two loops for double mutations that were capable of maintaining these tertiary interactions. Surprisingly, however, we found that most individual and double mutations do not reduce activity (*Figure 5*), and double mutants do not show positive epistasis (*Figure 5—figure supplement 1*). This indicates that the multiple interactions between the loops compensate for mutations that break a single interaction. It is interesting to note that the mutational robustness of these loops has been exploited in bioengineering applications, where insertion of an aptamer into one of the loops and randomization of the other allowed for the selection of synthetic riboswitches (*Townshend et al., 2015*). The identification of robust structural elements through high-throughput mutational data could be useful for identifying better targets for aptamer integration in other ribozymes.

We also find support for a two-nucleotide T1 pseudoknot in CPEB3 involving a non-canonical U-U base pair. While no crystal structure of the CPEB3 ribozyme has been solved, this U:U base pair has been confirmed and implicated as a magnesium binding site based on NMR and Tb3+ cleavage data (*Skilandat et al., 2016*). In our data, we find that G:C and C:G base pairs can support activity. We see negative epistasis for several pairs of mutations that result in 'mismatches' (A:G, A:A, and A:C) and positive or no epistasis for pairs of mutations that result in G:C and C:G base pairs, which all supports the formation of a second base pair in T1. We note that because the starting base pair is a U:U, the location of double mutants resulting in WC/GU pairs do not lie on the anti-diagonal in the heatmaps. Because a crystal structure of the CPEB3 ribozyme has not been solved, the CPEB3 data provides an example of how comprehensive mutational data can be useful for RNA with unknown structures.

## Evaluation of read depth and mutational coverage

The accuracy of our RA measurements depends on the number of reads we observe that map to each unique ribozyme sequence (read depth). Each reference ribozyme has a different nucleotide length resulting in different numbers of possible single and double mutants. In addition, the pooling

**Table 1.** Summary of the lengths of each self-cleaving ribozyme used in this study, the number of single and double mutants whose cleavage activity was analyzed, and the average fraction cleaved observed for all single and double mutants.

| Ribozyme name | Ribozyme length | Possible single Mmutants | Possible double mutants | Total mapped reads | Wild-type fraction cleaved | Single mutant average fraction cleaved | Double mutant average fraction cleaved |
|---|---|---|---|---|---|---|---|
| CPEB3 | 69 | 207 | 21,114 | 9,238,603 | 0.90 | 0.69 | 0.44 |
| HDV | 87 | 261 | 33,669 | 3,316,380 | 0.60 | 0.40 | 0.25 |
| Twister | 48 | 144 | 10,152 | 7,762,863 | 0.60 | 0.41 | 0.21 |
| Hairpin | 71 | 213 | 22,365 | 5,067,216 | 0.52 | 0.29 | 0.17 |
| Hammerhead | 45 | 135 | 8,910 | 8,054,498 | 0.34 | 0.27 | 0.19 |

of experimental replicates for sequencing does not result in equal mixtures of each replicate. In order to determine read depth, we mapped reads to the reference sequences and counted the number of reads that matched each ribozyme, while allowing for one or two mutations. We observed every single and double mutant for all ribozymes in each replicate, indicating 100% coverage of these mutant classes for all of our data sets. The distributions of observations for each single and double mutant of each ribozyme are shown in *Figure 6—figure supplement 3*. The HDV data showed the lowest depth, possibly because it is a larger ribozyme (87 nt), and fewer reads mapped to the single and double mutants (*Table 1*). Nevertheless, this analysis confirms that the data contain complete coverage of all single and double mutants and ample read depth for all five ribozymes.

## Epistasis plots are an informative approach to visualizing high-throughput activity data

Previous studies have reported comprehensive pairwise mutagenesis of ribozymes that provide interesting opportunities for comparison to the data presented here. For example, all pairwise mutations in a 42-nucleotide region of the same twister ribozyme were previously reported (*Kobori and Yokobayashi, 2016*). Compared to our experiments, these previous experiments used a later transcriptional time point (2 hr) and lower magnesium concentration (6 mM). They did not calculate epistasis, and reported the RA of all double mutants using heatmaps, inspiring the figures presented here. The results were highly similar, and the authors were able to identify paired regions in the data. The similarity between the results illustrates the reliability of this sequencing-based approach, which is promising for future data sharing and meta-analysis efforts. In another prior work, all pairwise mutations in the glmS ribozyme were analyzed using a custom-built fluorescent RNA array (*Andreasson et al., 2020*). The power of this approach is that they were able to monitor self-cleavage over short and long time scales, which enables differentiating both very slow and very fast self-cleaving variants. While the authors did not calculate pairwise epistasis, they reported RA heatmaps and also 'rescue effects' when the activity of a double mutant is sufficiently higher than the activity of a single mutant. This rescue analysis is very similar to positive epistasis, but only takes into account one mutation at a time. This

*Figure 5 continued*

are shaded blue, and negative epistasis is shaded red, with higher color intensity indicating a greater magnitude of epistasis. Catalytic residues are indicated by stars along the axes. (**B**) Secondary structure of the hammerhead ribozyme used in this study. Each nucleotide is shaded to indicate the average relative cleavage activity of all single mutations at that position. (**C**) Distributions of epistasis values in the paired regions of the hammerhead ribozyme. Data were categorized as double mutations that result in two mismatches (2 Mismatch), a single mismatch (1 Mismatch), or no mismatches because of a new Watson-Crick base pair or GU wobble results (WC/GU). (**D**) Crystal structure of a hammerhead ribozyme (3ZD5) with C20 and G25 indicated (orange) and hydrogen bonds between the nucleotides shown as yellow dashed lines.

The online version of this article includes the following figure supplement(s) for figure 5:

**Figure supplement 1.** Distribution of pairwise epistasis observed between the loops of P1 and P2 in the hammerhead ribozyme.

*Figure 6 continued*

The online version of this article includes the following figure supplement(s) for figure 6:

**Figure supplement 1.** Correlation between mutational effects and RNA helix stability.

**Figure supplement 2.** Relative activity (RA) values for sequences with mutations at catalytic and non-catalytic nucleotide positions.

**Figure supplement 3.** Histogram of the distributions of read counts (read depth) for the single and double mutants matching to each ribozyme analyzed in this study (CPEB3, HDV, twister, hairpin, and hammerhead).

analysis was also able to identify many of the know base-pair interactions and some tertiary contacts in the glmS ribozyme. In addition, they were able to observe some minor secondary structure rearrangement, where mutations in some nucleotides were able to rescue neighboring nucleotides by shifting the base-pairing slightly. The pairwise epistasis analysis presented here adds an additional approach to extract information from such high-throughput sequencing-based analysis of self-cleaving ribozymes. Unlike the rescue analysis, which can only identify positive interactions, the ability to detect negative epistatic interactions may help further identify structurally important regions for RNA sequence design and engineering efforts. It is possible that all of these analysis approaches could be used for RNA functions other than self-cleavage, if they can be detected by high-throughput sequencing. This could include ribozyme activities that can be enriched by in vitro selections (*Pressman et al., 2019*), or mutations in natural RNA molecules that affect growth rates (*Li et al., 2016*).

## Conclusion

We have determined the RA for all single and double mutants of five self-cleaving ribozymes and use this data to calculate epistasis for all possible pairs of nucleotides. The data was collected under identical co-transcriptional conditions, facilitating direct comparison of the data sets. The data revealed signatures of structural elements including paired regions and non-canonical structures. In addition, the comprehensiveness of the double mutants enabled identification of catalytic residues. Recently, there has been significant progress toward predicting RNA structures from sequence using machine learning approaches (*Calonaci et al., 2020*; *Townshend et al., 2021*; *Zhao et al., 2021*). The machine learning models are typically trained on structural biology data from x-ray crystallography, chemical probing (SHAPE), and natural sequence conservation. Self-cleaving ribozymes have been central to this effort. Our approach is similar to SHAPE in that it can be obtained with common lab equipment and commercially available reagents. The activity data presented provides information similar to natural sequence conservation, except that it provides quantitative effects of mutations, not just frequency. For example, secondary structures have been predicted based on comparative sequence analysis by identifying covarying nucleotide positions in homologous RNA sequences. These approaches are important because they do not require any experimental evaluation of sequences. However, this 'comparative approach' may not be able to identify important nucleotides or structural elements other than canonical base pairs. We hope that the activity-based data presented here will provide information not present in these other training data sets and help advance computational predictions.

## Materials and methods
### Mutational library design and preparation of self-cleaving ribozymes

Single-stranded DNA molecules used as templates for in vitro transcription were synthesized as described previously (*Kobori and Yokobayashi, 2016*), using doped oligos containing 97% of the base of the reference sequence and 1% of the three other remaining bases at each position (Keck Oligo Synthesis Resource, Yale). A constant structured RNA cassette was appended to the template sequences to provide a reverse transcription primer binding site (*Wilkinson et al., 2006*). The ssDNA library was made double-stranded to allow for T7 transcription via low-cycle PCR using Taq DNA polymerase.

### Co-transcriptional self-cleavage assay

The co-transcriptional self-cleavage reactions were carried out in triplicate by combining 20 µL 10× T7 transcription buffer (500 µL 1 M Tris pH 7.5, 50 µL 1 M DTT, 20 µL 1 M Spermidine, 150 µL 1 M MgCl2,

280 µL RNase-free water), 4 µL rNTP (25 mM, NEB, Ipswich, Ma), 8 µL T7 RNA Polymerase-Plus enzyme mix (1600 U, Invitrogen, Waltham, MA), 160 µL nuclease-free water, and 8 µL of double-stranded DNA template (4 pmol, 0.5 µM PCR product) at 37°C for 30 min. The transcription and co-transcription self-cleavage reactions were quenched by adding 60 µL of 50 mM EDTA. The resulting RNA was purified and concentrated using Direct-zol RNA MicroPrep Kit with TRI-Reagent (Zymo Research, Irvine, CA), and eluted in 7 µL nuclease-free water. Concentrations were determined via absorbance at 260 nm (ThermoFisher NanoDrop, Waltham, MA), and normalized to 5 µM.

## Reverse transcription and Illumina indexing PCR

Reverse transcription was carried out using a 5′RACE protocol using phased template switching oligo's (TSO1-4, *Supplementary file 1*) as described previously (*Bendixsen et al., 2020*). Briefly, reverse transcription reactions used 5 pmol RNA and 20 pmol of reverse transcription primer in a volume of 10 µL. RNA and primer were heated to 72°C for 3 min and cooled on ice. Reverse transcription was initiated by adding 4 µL SMARTScribe 5× First-Strand Buffer (TaKaRa, San Jose, CA), 2 µL dNTP (10 mM), 2 µL DTT (20 mM), 2 µL phased template switching oligo mix (10 µM), and 2 µL SMARTScribe Reverse Transcriptase (200 units, TaKaRa). Four different template switching oligos (TSO 1–4) with different lengths and nucleotide compositions were used such that the first nucleotides read during sequencing are a balance of all four nucleotides, and the ribozymes are sequenced in four different frames relative to the primer. The mixture was incubated at 42°C for 90 min and the reaction was stopped by heating to 72°C for 15 min. The resulting cDNA was purified on a silica-based column (DCC-5, Zymo Research) and eluted into 7 µL water. Illumina adapter sequences and indexes were added using high-fidelity PCR. A unique index combination was assigned to each ribozyme and for each replicate. The PCR reaction contained 3 µL purified cDNA, 12.5 µL KAPA HiFi HotStart ReadyMix (2×, KAPA Biosystems, Wilmington, MA), 2.5 µL forward, 2.5 µL reverse primer (Illumina Nextera Index Kit), and 5 µL water. Several cycles of PCR were examined using gel electrophoresis and a PCR cycle was chosen that was still in logarithmic amplification, prior to saturation. Each PCR cycle consisted of 98°C for 10 s, 63°C for 30 s, and 72°C for 30 s. PCR DNA was purified on silica-based columns (DCC-5, Zymo Research) and eluted in 22.5 µL water. The final product was then verified using gel electrophoresis.

## High-throughput sequencing

The indexed PCR products for all replicates were pooled together at equimolar concentrations based on absorbance at 260 nm. Paired-end sequencing reads were obtained for the pooled libraries using an Illumina HiSeq 4000 (Genomics and Cell Characterization Core Facility, University of Oregon).

## Sequencing data analysis

Paired-end sequencing reads were joined using FLASh, allowing 'outies' due to overlapping reads. The joined sequencing reads were analyzed using custom Julia scripts available at https://gitlab.com/bsu/biocompute-public/mut_12 (*Beck, 2023*). Our analysis implemented a sequence-length sliding window to screen for double mutant variants of a reference ribozyme. Nucleotide identities for each mutant were identified and then counted as either cleaved or uncleaved based on the presence or absence of the 5′-cleavage product sequence. The RA was calculated as previously described (*Kobori and Yokobayashi, 2016*). Briefly, a FC was calculated for each genotype in each replicate as $FC=N_{clv}/(N_{clv}+N_{unclv})$. This value was normalized to the reference/wild-type FC as $RA=FC/FC_{wt}$. The RA values were averaged across the three replicates and then plotted as a heatmap. Epistasis interactions for each double mutant (i, j) were quantified as previously described (*Bendixsen et al., 2017*), where $\text{Epistasis}\,(\varepsilon) = \log\frac{RA_{i,j}}{(RA_i)\,(RA_j)}$ . In order to eliminate false positive detection of epistasis interactions, values were filtered to eliminate instances where the difference between the double and any of the single mutants was less than 1–3σ of the overall distribution of differences between the single and double mutant relative activities. Values greater than 1 indicate positive epistasis, and values less than 0 indicate negative epistasis. Mann-Whitney U test was used to determine the probability that epistasis or activity values of different structural elements were from the same distribution.

### Correlation of thermodynamic stability of paired regions and observed mutational effects

Each base-paired region was split into two separate RNA sequences containing only the nucleotides involved in base-pairing, omitting nucleotides belonging to stem loops. Complex formation between each pair of strands at was analyzed in Nupack using Serrra and Turner RNA energy parameters in order to obtain minimum free energy values for each paired region (37°C, [1 µM]). Using custom Julia scripts, the median RA for single mutations to each paired region was plotted as a function of the calculated free energy and a Pearson correlation coefficient was calculated.

## Acknowledgements

The authors acknowledge funding from the National Science Foundation (EH, Grant numbers OIA-1738865 and OIA-1826801), the National Aeronautics and Space Administration (EH, Grant number 80NSSC17K0738), and the Human Frontier Science Program (EH, Ref no.: RGY0077/2019).

## Additional information

### Funding

| Funder | Grant reference number | Author |
| --- | --- | --- |
| National Science Foundation | OIA-1738865 | Eric J Hayden |
| National Science Foundation | OIA-1826801 | Eric J Hayden |
| National Aeronautics and Space Administration | 80NSSC17K0738 | Eric J Hayden |
| Human Frontier Science Program | RGY0077/2019 | Eric J Hayden |

The funders had no role in study design, data collection and interpretation, or the decision to submit the work for publication.

### Author contributions

Jessica M Roberts, Conceptualization, Formal analysis, Supervision, Validation, Investigation, Visualization, Methodology, Writing – original draft, Writing – review and editing; James D Beck, Data curation, Software, Formal analysis, Visualization, Methodology, Writing – review and editing; Tanner B Pollock, Validation, Investigation; Devin P Bendixsen, Conceptualization, Investigation, Methodology, Writing – review and editing; Eric J Hayden, Conceptualization, Resources, Supervision, Funding acquisition, Writing – original draft, Writing – review and editing

### Author ORCIDs

Jessica M Roberts http://orcid.org/0000-0003-3164-7256
James D Beck http://orcid.org/0000-0002-9086-2653
Devin P Bendixsen http://orcid.org/0000-0003-0831-7646
Eric J Hayden http://orcid.org/0000-0001-6078-5418

### Decision letter and Author response

Decision letter https://doi.org/10.7554/eLife.80360.sa1
Author response https://doi.org/10.7554/eLife.80360.sa2

## Additional files

### Supplementary files
• Supplementary file 1. Oligonucleotides used in this study.

• MDAR checklist

## Data availability

Sequencing reads in FastQ format are available at ENA (PRJEB52899 and PRJEB51631). Sequences, activity data, and computer code is available at GitLab (https://gitlab.com/bsu/biocompute-public/mut_12 copy archived at swh:1:rev:7be31e58784ab6ffe5f15790ebbda98279af84d2).

The following previously published datasets were used:

| Author(s) | Year | Dataset title | Dataset URL | Database and Identifier |
|---|---|---|---|---|
| Roberts JM, Beck JD, Pollock TB, Bendixsen DP, Hayden EJ | 2022 | RNA sequence to structure analysis from comprehensive pairwise mutagenesis of multiple self-cleaving ribozymes | https://www.ebi.ac.uk/ena/browser/view/PRJEB52899 | European Nucleotide Archive, PRJEB52899 |
| Beck JD, Roberts JM, Kitzhaber JM, Trapp A, Serro E, Spezzano F, Hayden EJ | 2022 | Predicting higher-order mutational effects in an RNA enzyme by machine learning of high-throughput experimental data | https://www.ebi.ac.uk/ena/browser/view/PRJEB51631 | European Nucleotide Archive, PRJEB51631 |

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
