## [Editor Report]

This is a valuable study that provides compelling evidence for important nucleotides in five self-cleaving ribozymes. Epistasis analyses are novel in this field.

---

## [Decision Letter]

**Decision letter after peer review:**

Thank you for submitting your article "RNA sequence to structure analysis from comprehensive pairwise mutagenesis of multiple self-cleaving ribozymes" for consideration by *eLife*. Your article has been reviewed by 3 peer reviewers, and the evaluation has been overseen by Timothy Nilsen as Reviewing Editor and James Manley as the Senior Editor. The following individuals involved in the review of your submission have agreed to reveal their identity: Philip Bevilacqua (Reviewer #1); Benoît Masquida (Reviewer #2).

All the reviewers felt that the work was, in principle, suitable for publication in *eLife*. Nevertheless, the reviewers felt that the data was under-interpreted and that analyses from a 3D perspective would improve the paper significantly. Please see the reviews below. In this case, any revised manuscript will be subject to rereview.

*Reviewer #1 (Recommendations for the authors):*

1. p8. The longer size of HDV does not explain the fewer reads for it because it is only slightly longer than the other ribozymes. What about the stability of the HDV ribozyme and RT not being able to read through it?

2. p8. What is the minimum number of reads per single mutant in HDV? double mutants? For the latter, the average is only 50 (Figure S1) and the minimum appears to be ~10 reads. Can reliable data be attained with so few reads? What is the statistical significance for low read singles and double mutations?

3. p8. more precise language is needed. Suggestions below. These words do not have to be used but some should be provided to guide the reader. "We plotted the relative activity value as heat maps (Figures 1-5 a large plot, shows only blue color)." "We then used this data to calculate epistasis between pairs of mutations (Figures 1-5 insets red to blue colors)".

4. p8. "many paired regions showed an anti-diagonal line of high activity double mutant variants with strong positive epistasis". It would be helpful to dissect this anti-diagonal into two different distributions: WC+wobbles and mismatches. In other words, there are off-diagonal elements within the anti-diagonal square that are meaningful. This should provide an interesting sub-anativealysis in panel C; specifically, it will allow a look at double mutants that lead to the loss of a single base pair (off-diagonal elements within the anti-diagonal square) vs. a loss of two pairs (off-diagonal squares).

5. Overall fraction cleaved varies wildly between the five ribozymes, from 0.19 to 0.68 (Table 1). It would be helpful to know what the fraction cleaved was for the wild-type ribozymes that emerged from the deep sequencing versus just making the wild-type ribozyme alone and measuring the bulk fraction cleaved from a radiolabeled experiment. This could help the authors discuss how sensitive or robust a given ribozyme is to mutations and then speculate why.

6. Why is panel A different colors for different ribozymes? This seems unnecessary and random (e.g. two of them are blue).

7. p9. Not only were the epistasis values for on- anti-diagonal "consistently more positive than two mutations in positions that are not directly base-paired (off-diagonal)" the latter were often negative (at least the mean was), which should be stated. i.e. "more positive" implies both distributions were positive.

8. Figure 4. The inset for LA/LA is not positioned correctly on the main figure. A28-A31 are should be left-shifted. Moreover, epistasis between LA and LB could not be judged, as the authors would like the reader to do on p12. The authors should provide an LA/LB inset for the reader to look at. Additionally, the authors would like the reader to think about interactions of 1 and 46 but don't draw the pair in panel B, making this hard to visualize. The "positive epistasis" for G42U and A64G is missing in the inset which is entirely white for this square. And the A47:G57 showing "positive epistasis for double mutants that result in an AU base pair" doesn't make sense because this can happen with a single mutant at G57U.

9. p14. The authors give details on epistasis between positions 20 and 25 but nothing is shown in the off-diagonals and this data cannot be had on the main diagram in A.

10. p10. The discussion that "differences between epistasis in short and long base-paired regions suggests that the thermodynamic stability of each paired region is important for the observed activity" is clearly supported by the data but should be explained. Short regions cannot withstand the loss of one and especially two base pairs because they are short and once broken cannot be broken again. Bevilacqua and Herschlag have discussed this and could be consulted and referenced. 1. Moody, E. M. and Bevilacqua, P. C. (2003). Folding a stable DNA motif involves a highly cooperative network of interactions. J. Am. Chem. Soc. 125, 16285-16293. 2. Kraut, D. A., Carroll, K. S. and Herschlag, D. (2003). Challenges in enzyme mechanism and energetics. Annu. Rev. Biochem. 72, 517-571.

11. Methods. The authors did reverse transcription. From Table S1, it appears that extra bases were appended to the 3'end of each ribozyme to provide an RT primer binding site [it appears this way from the first unbolded bases listed at the 5'-end of the template in Table S1]. If so how do those bases affect the fraction cleaved of the wild-type? And then the mutants? See comment 5 above. Also, I was unable to understand how the sequence of the RT primer in Table S1 works with the templates.

12. In the epistasis equation, which should be numbered, what are the authors taking the log of? RA? Shouldn't there be a plus sign between the two terms in the denominator? For e.g. assume additivity RAi = RAj = 10^-1, and RAij = 10^-2, we need a plus sign to have this come out to unity; as it would come out to -2. Also, how does it follow that negative epistasis is less than 0? Shouldn't negative be >1 and positive be <1?

13. We could see some trend in the length of paired regions and the intensity of the epistasis effect, but it was hard to tell whether the negative correlation between the median deleterious effects of single mutations and the minimum free energy of the paired regions was significant from the plot in Supplementary Figure 3 and the given Pearson Correlation = -0.53. Also, it may be good to address and explain the difference in the distribution of epistasis value of CPEB3 P1, P2, and P4, which all have 7 base pairs but are very different in the distribution of epistasis value.

*Reviewer #2 (Recommendations for the authors):*

Regarding the CPEB3 ribozyme, an open question is about the role of the U21-U42 base pair. Figure 1 indicates that positive epistasis has been measured for some sequence combinations. It would be very sound to frame this region of the heat map together with the T1 interaction to discuss the heuristic power of the presented approach since the CPEB3 ribozyme is the only ribozyme studied in this manuscript for which no crystal structure has yet been made available.

*Reviewer #3 (Recommendations for the authors):*

1) For example, the way the ribozymes have been randomized is already described in Kobori and Yokobayashi (2016). On the other side, it would be necessary to provide more experimental information on the way the template switching reverse transcription is performed. This is not well explained as there is no clear explanation for the usage of the TSO1-4 oligos… The authors should rewrite the experimental method section of their article so that anybody who wants to use this approach could do it without struggling…

2) While some of this 3D information is seldom mentioned in the text, there is no easy way for the reader to find this information in the data provided. A figure exemplifying some of the key 3D interactions of these ribozymes would be most useful.

For example, it would be nice to have more structural details by eventually showing some of the results within the context of the 3D structures of each ribozyme.

For instance, the Watson-Crick base pair interaction G1C-C46G between LA/LB in the hairpin ribozyme could be shown… The same thing with the base pair C20G-G25C in the hammerhead ribozyme. In fact, the 2D structure of the hammerhead ribozyme could be improved as it does not correspond to the active form.

The authors should also provide more data information (as well as supportive figures) about the tertiary interactions that involve non-canonical base pairs and that show positive epistasis… After all, this is a piece of information that has not yet been obtained before for these ribozymes as it cannot be easily obtained by other approaches.

3) The authors could have certainly enhanced dramatically the scope of their article by trying to validate the structure of a self-cleaving ribozyme for which the 3D structure is not known yet. This would have provided a clear test for their approach and would have enhanced dramatically their claim that it could complement chemical and enzymatic probing.

---

## [Author Response]

Reviewer #1 (Recommendations for the authors):1. p8. The longer size of HDV does not explain the fewer reads for it because it is only slightly longer than the other ribozymes. What about the stability of the HDV ribozyme and RT not being able to read through it?

There are several factors that may have resulted in fewer reads for HDV. We did normalize the amount of RNA going into the RT reaction, but we did not quantify the amount of resulting cDNA going into the indexing PCR reaction. As you mentioned, the stability of HDV could have influenced this, but we are lacking any direct information. We used Fragment Analyzer data to quantify the amount of correct sized PCR DNA before pooling the samples, so some unequal mixing is possible, but unlikely. A very probable cause is that all samples were sequenced on a single flow cell. It is reported by Illumina that shorter sequences cluster more efficiently, likely due to biased bridge amplification on the flow cell, which would have favored the shorter ribozyme sequences. Finally, the doped oligo synthesis used to generate the library using a 3% mutation rate results in a mean number of mutations of 2.6 for HDV, whereas we would expect a mean number of mutations closer to 2 for the other ribozymes. This results in the library containing more sequences with three or more mutations, which were excluded from our analysis. The combined effects of lower numbers of single and double mutants, clustering bias and possible RT efficiency are all probably contributing to the lower coverage.

2. p8. What is the minimum number of reads per single mutant in HDV? double mutants? For the latter, the average is only 50 (Figure S1) and the minimum appears to be ~10 reads. Can reliable data be attained with so few reads? What is the statistical significance for low read singles and double mutations?

The minimum number of reads for single mutants for HDV is 1945, and the minimum number of reads for double mutants is 4. We acknowledge that sequences that have very few reads have lower confidence in the calculated fraction cleaved and relative activity. In fact, we found a weak correlation between the standard deviation of the fraction cleaved between replicates and the number of reads (pearson = -0.02). The lowest abundance reads have the least confidence in the mean relative activity, which we agree could affect some of our epistasis calculations because all of the lowest abundance reads are double mutants. We are confident in the single mutants, but some of the double mutants could be reported higher or lower due to low abundance. To address this concern, we inspected the data and found that there are only 36 HDV sequences that have 10 or fewer reads (0.1% of 33,669 double mutant genotypes). We therefore looked at the structural location of each pair of mutations. For 34 out of the 36 each of the two mutations were located in different structural regions, and therefore do not contribute to our observations of activity or epistasis *within* structural elements, which is the main analysis in the text. There were only two instances of low abundance reads where both of the mutations fell within a single structural element: (1) U12C and A16G are both in the same side of HDV P2, and do not form a base pair. The activity of the double mutant sequence is used to calculate one epistasis data point that is included in the “off diagonal” epistasis distribution presented in the figure. (2) U20C and U27G is a double mutant where both mutations reside within the loop L3. The activity of the double mutant is used to calculate one epistasis data point in the Terminal loop data used for comparison to the hairpin loop data in Figure 4D. Taken together we feel that the low abundance reads in the HDV data set do not contribute in any significant way to the analysis and conclusions in the manuscript.

It is also important to note that we took another precaution in all of our epistasis analysis. We were concerned that very low activity single mutants could lead to positive epistasis even when the double mutant had only slightly higher activity, but was still very low activity. We felt that this minor rescue effects were likely of little structural relevance. In our analysis, all epistasis values were evaluated based on the differences between single and double mutants, and likely “false positives” were filtered from our plots. To do this, we determined the distribution of all difference between double mutants and their comprising single mutations. We calculated the standard deviation of this distribution. We set epistasis to zero for instances where the difference between the double mutant and either of the single mutants was less than 1-3stdev of the overall distribution. This filtering approach has also been used by others referenced below.

1. Andreasson JOL, Savinov A, Block SM, Greenleaf WJ (2020) Comprehensive sequence-to-function mapping of cofactor-dependent RNA catalysis in the glmS ribozyme. *Nature Communications*, 11(1):1663. https://doi.org/10.1038/s41467-020-15540-1

3. p8. more precise language is needed. Suggestions below. These words do not have to be used but some should be provided to guide the reader."We plotted the relative activity value as heat maps (Figures 1-5 a large plot, shows only blue color).""We then used this data to calculate epistasis between pairs of mutations (figures 1-5 insets red to blue colors)".

Thank you for this suggestion, we agree more precise language would help the reader discern between the RA plot and the epistasis insets. We have edited the text on page 8 to state “(Figures 1-5 A, large plot)”, and “(Figures 1-5, insets, red to blue plots)”.

4. p8. "many paired regions showed an anti-diagonal line of high activity double mutant variants with strong positive epistasis". It would be helpful to dissect this anti-diagonal into two different distributions: WC+wobbles and mismatches. In other words, there are off-diagonal elements within the anti-diagonal square that are meaningful. This should provide an interesting sub-anativealysis in panel C; specifically, it will allow a look at double mutants that lead to the loss of a single base pair (off-diagonal elements within the anti-diagonal square) vs. a loss of two pairs (off-diagonal squares).

Thank you for this suggestion. We performed the suggested analysis and did find there was a significant difference between anti-diagonal elements which formed a WC base pair or GU wobble, from those that broke a single base pair, and these were also significantly different from off-diagonal mutations that broke 2 base pairs. We have updated panel C in Figures 1-5 with these distributions shown as violin plots, and have also revised the text on page 9 to reflect this new analysis.

5. Overall fraction cleaved varies wildly between the five ribozymes, from 0.19 to 0.68 (Table 1). It would be helpful to know what the fraction cleaved was for the wild-type ribozymes that emerged from the deep sequencing versus just making the wild-type ribozyme alone and measuring the bulk fraction cleaved from a radiolabeled experiment. This could help the authors discuss how sensitive or robust a given ribozyme is to mutations and then speculate why.

Thank you for this suggestion. We have updated Table 1 to indicate the fraction cleaved that we observed for the wild-type, for single, and for double mutants for each ribozyme.

6. Why is panel A different colors for different ribozymes? This seems unnecessary and random (e.g. two of them are blue).

Yes, we agree this is arbitrary. However, we would prefer to leave them colored as they are because they coordinate with several supplemental figures, and we believe it makes the data sets easier to differentiate. We chose color blind friendly colors, so we used two different blues that are easy to distinguish by color blind individuals.

7. p9. Not only were the epistasis values for on- anti-diagonal "consistently more positive than two mutations in positions that are not directly base-paired (off-diagonal)" the latter were often negative (at least the mean was), which should be stated. i.e. "more positive" implies both distributions were positive.

We agree that this language could be improved for clarity and to emphasize our observation of prevalent negative epistasis ‘off-diagonal’. We edited the text on page 9 to clarify these points.

8. Figure 4. The inset for LA/LA is not positioned correctly on the main figure. A28-A31 are should be left-shifted. Moreover, epistasis between LA and LB could not be judged, as the authors would like the reader to do on p12. The authors should provide an LA/LB inset for the reader to look at. Additionally, the authors would like the reader to think about interactions of 1 and 46 but don't draw the pair in panel B, making this hard to visualize. The "positive epistasis" for G42U and A64G is missing in the inset which is entirely white for this square. And the A47:G57 showing "positive epistasis for double mutants that result in an AU base pair" doesn't make sense because this can happen with a single mutant at G57U.

Thank you, we corrected the placement of the box for the LA/LA inset on the hairpin figure.

Moreover, epistasis between LA and LB could not be judged, as the authors would like the reader to do on p12. The authors should provide an LA/LB inset for the reader to look at. Additionally, the authors would like the reader to think about interactions of 1 and 46 but don't draw the pair in panel B, making this hard to visualize.

In order to help readers visualize the interaction between G1 and C46, we have drawn a dashed line between these positions on the secondary structure in Figure 4. In addition, we have generated a new supplementary figure (S.Figure 5) that we think better highlights our epistasis data as it corresponds to the loop-loop interactions in hairpin. This figure contains a secondary structure of the loopA-loopB interactions that depicts which bases have been previously shown to interact (panel A), a crystal structure rendering of the loop loop interaction (panel B), as well as the LoopA-LoopB epistasis insets (panel C). We think that the linking of Supplemental figures to main figures in *eLife* should make this easily accessible for the reader.

The "positive epistasis" for G42U and A64G is missing in the inset which is entirely white for this square. And the A47:G57 showing "positive epistasis for double mutants that result in an AU base pair" doesn't make sense because this can happen with a single mutant at G57U.

You are correct that there were errors in some of our examples of epistasis in these loops. Thank you for catching these mistakes! We have closely re-inspected epistasis in these areas and have updated the description of the results on page 13 of the manuscript. Regarding the A47:G57 mutations that result in an AU basepair, you are correct that the A47:G57U would only be a single mutant, and would not exhibit epistasis. We have edited this to simply state we see positive epistasis for the A47G:G57U mutations, and have removed the language stating that this forms an AU basepair.

9. p14. The authors give details on epistasis between positions 20 and 25 but nothing is shown in the off-diagonals and this data cannot be had on the main diagram in A.

We agree that this data should be shown in the main figure for hammerhead. We have added an inset to show epistasis values for the nucleotides in the CUGA motif that contains this interaction. We have also added a panel (D) to the figure showing a crystal structure with this base pair interaction highlighted.

10. p10. The discussion that "differences between epistasis in short and long base-paired regions suggests that the thermodynamic stability of each paired region is important for the observed activity" is clearly supported by the data but should be explained. Short regions cannot withstand the loss of one and especially two base pairs because they are short and once broken cannot be broken again. Bevilacqua and Herschlag have discussed this and could be consulted and referenced. 1. Moody, E. M. and Bevilacqua, P. C. (2003). Folding a stable DNA motif involves a highly cooperative network of interactions. J. Am. Chem. Soc. 125, 16285-16293. 2. Kraut, D. A., Carroll, K. S. and Herschlag, D. (2003). Challenges in enzyme mechanism and energetics. Annu. Rev. Biochem. 72, 517-571.

Thank you for this suggestion, we added those references and your suggested explanation. We agree that for regions that are less thermodynamically stable, mutational effects seem more extreme. Our data shows that this does not result in more negative epistasis, but rather a shift towards more positive epistasis for compensatory double mutations, and that appears to be responsible for the large differences in the epistasis distributions for these areas. We also clarified the text to explain this.

11. Methods. The authors did reverse transcription. From Table S1, it appears that extra bases were appended to the 3'end of each ribozyme to provide an RT primer binding site [it appears this way from the first unbolded bases listed at the 5'-end of the template in Table S1]. If so how do those bases affect the fraction cleaved of the wild-type? And then the mutants? See comment 5 above. Also, I was unable to understand how the sequence of the RT primer in Table S1 works with the templates.

Thank you for your close inspection of the oligos used. Our templates did contain additional bases to provide an RT primer site. The sequence used was developed by others as a primer binding site for RNA SHAPE experiments. It was reported in the literature that the sequence did not interfere with the folding of several different RNA molecules. To prevent structural interference, the sequence is designed to have 2 internal hairpins each stabilized by a UUCG tetraloop which are separated by short dinucleotide linkers to reduce stacking of the hairpins on the internal RNA structure. Because it is on the 3’-end of the RNA, and we are studying co-transcriptional self-cleavage, the ribozyme structure has a kinetic advantage as well because it is transcribed first and has the opportunity to fold before the primer binding site is transcribed. We do not have any direct information on how this sequence may affect the fraction cleaved of every mutant sequence, but have observed high observed rates of self-cleavage for the wild type (see table 1). In previous publications with the CPEB3 ribozymes, we observed no difference in activity with and without the primer binding site (unpublished data). We have added the following reference to the manuscript because it describes the design and effectiveness of the primer binding sequence in SHAPE experiments:

Wilkinson KA, Merino EJ, Weeks KM. 2006. Selective 2′ -hydroxyl acylation analyzed by primer extension (SHAPE): quantitative RNA structure analysis at single nucleotide resolution. Nat Protoc 1: 1610–1616. doi:10.1038/nprot.2006.249 Wu L, Wen C, Qin Y, Yin H, Tu Q, Van N

Also, I was unable to understand how the sequence of the RT primer in Table S1 works with the templates.

The RT primer listed in the table was an erroneous duplication of our TSO4 sequence. The sequence was replaced with the correct RT primer sequence. Thank you for finding this mistake.

12. In the epistasis equation, which should be numbered, what are the authors taking the log of? RA? Shouldn't there be a plus sign between the two terms in the denominator? For e.g. assume additivity RAi = RAj = 10^-1, and RAij = 10^-2, we need a plus sign to have this come out to unity; as it would come out to -2. Also, how does it follow that negative epistasis is less than 0? Shouldn't negative be >1 and positive be <1?

Thank you for catching this, we made a mistake when writing the equation we used to calculate epistasis in the manuscript. We have corrected it in the manuscript and also double checked our code that we used to calculate epistasis. We found that the code was correct, so this was just a typo that is now corrected. Because there are only a few equations reported and none are referenced elsewhere in the manuscript, we did not number them. We will defer to the editorial preference of the journal.

13. We could see some trend in the length of paired regions and the intensity of the epistasis effect, but it was hard to tell whether the negative correlation between the median deleterious effects of single mutations and the minimum free energy of the paired regions was significant from the plot in Supplementary Figure 3 and the given Pearson Correlation = -0.53. Also, it may be good to address and explain the difference in the distribution of epistasis value of CPEB3 P1, P2, and P4, which all have 7 base pairs but are very different in the distribution of epistasis value.

We have added a trendline to the plot, and have added the p = value of 0.029 to show significance. CPEB3 P1 appears to be more sensitive to the first mutation, likely because of proximity to the cleavage site. We have added this to the discussion of thermodynamic stability starting on p. 10.

Reviewer #2 (Recommendations for the authors):Regarding the CPEB3 ribozyme, an open question is about the role of the U21-U42 base pair. Figure 1 indicates that positive epistasis has been measured for some sequence combinations. It would be very sound to frame this region of the heat map together with the T1 interaction to discuss the heuristic power of the presented approach since the CPEB3 ribozyme is the only ribozyme studied in this manuscript for which no crystal structure has yet been made available.

Thank you for this suggestion. We have taken a closer look at our data for single and double mutants involving the analogous U21 and U38 for our construct used. We believe that our data does support the presence of this flexible non-WC base pair interaction as described in Skilindat, et al. 2016. For nearly all double mutations involving a U->A mutation, we observe negative epistasis. This could be caused by single U->A mutations forming a stable AU basepair, and exhibiting an increase (or no decrease) in RA for those single mutants. Therefore, double mutations involving a U->A mutation that do not form a BP will disrupt this favorable single mutant interaction and would be expected to exhibit negative epistasis. Additionally, we do see weak positive epistasis for the U21C:U38G double mutant, which could hypothetically form a GC basepair. While we did not observe positive epistasis for the U21G:U38C double mutant, we believe this could be influenced by the effects of single mutations going through a GU wobble.

Worth noting however, is that this basepair interaction is not as sensitive to single and double mutations as we would expect for such a short paired region. Most single and double mutations to either of these positions do not result in a significant reduction in self-cleavage activity. In fact, the most sensitive single mutation is U38G, which we would expect to be a favorable mutation if it formed a GU wobble with U21. Skilindat, et al. 2016 found that U38 is a site of Mg^2+^ binding. Our data suggests that this Mg^2+^ binding interaction is not necessarily nucleotide specific, as U38G is the only deleterious single mutation at this position.

We have expanded the inset for T1 for CPEB3 to show the epistatic effects of all double mutations between these two positions, and have indicated the presence of that interaction on the secondary structure in figure 1. We have also added a discussion to the text on page 15.

Reviewer #3 (Recommendations for the authors):1) For example, the way the ribozymes have been randomized is already described in Kobori and Yokobayashi (2016). On the other side, it would be necessary to provide more experimental information on the way the template switching reverse transcription is performed. This is not well explained as there is no clear explanation for the usage of the TSO1-4 oligos… The authors should rewrite the experimental method section of their article so that anybody who wants to use this approach could do it without struggling…

Thank you for your comment, we have added the appropriate Kobori and Yokobayashi reference for our doped oligo synthesis library design to our methods section.

Our lab has previously published a manuscript detailing the template switching reverse transcription using phased oligos methods in the RNA journal, we have made sure to add a clear citation that can be referenced by those wishing for more detailed information to use this approach.

Bendixsen DP, Roberts JM, Townshend B, Hayden EJ. 2020. Phased nucleotide inserts for sequencing low-diversity RNA samples from in vitro selection experiments. *RNA* 26:1060–1068. doi:10.1261/rna.072413.119

2) While some of this 3D information is seldom mentioned in the text, there is no easy way for the reader to find this information in the data provided. A figure exemplifying some of the key 3D interactions of these ribozymes would be most useful.For example, it would be nice to have more structural details by eventually showing some of the results within the context of the 3D structures of each ribozyme.For instance, the Watson-Crick base pair interaction G1C-C46G between LA/LB in the hairpin ribozyme could be shown.

Thank you for this suggestion. We agreed that our manuscript would be improved by the incorporation of 3D figures exemplifying the key 3D interactions that we discussed. We have added a new supplementary figure 5 which highlights the 3D interactions within and between the internal loops of hairpin. This figure includes a secondary structure depiction of these specific interactions, as well as a crystal structure rendering that shows the loop-loop interactions and highlights the G1-C46 basepair. The same thing with the base pair C20G-G25C in the hammerhead ribozyme. We have also added a new panel D to the hammerhead figure (5), which shows a crystal structure for hammerhead highlighting the C20G-G25C basepair interaction.

In fact, the 2D structure of the hammerhead ribozyme could be improved as it does not correspond to the active form…

We are not aware of a more accurate 2D depiction of the hammerhead ribozyme, as our version is shown in the form that allows for the tertiary contact between L1 and L2.

The authors should also provide more data information (as well as supportive figures) about the tertiary interactions that involve non-canonical base pairs and that show positive epistasis… After all, this is a piece of information that has not yet been obtained before for these ribozymes as it cannot be easily obtained by other approaches…

We have added all of our epistasis data for double mutants to the new supplementary figure 5 highlighting the loop-loop interaction in hairpin. We believe this will allow readers to explore instances of positive epistasis measured for the tertiary interactions involving non-canonical pairs in the hairpin loops.

3) The authors could have certainly enhanced dramatically the scope of their article by trying to validate the structure of a self-cleaving ribozyme for which the 3D structure is not known yet. This would have provided a clear test for their approach and would have enhanced dramatically their claim that it could complement chemical and enzymatic probing.

Thank you for this comment. We do agree this would have strengthened the manuscript, but is beyond the scope of the work contained herein.